# Fracture System and Rock-Mass Characterization by Borehole Camera Surveying: Application in Dimension Stone Investigations in Geologically Complex Structures

**Ivica Pavičić [1,*]** , **Ivo Galić [1]** , **Mišo Kucelj [2]** and **Ivan Dragičević [1]**

[1]  Faculty of Mining, Geology and Petroleum Engineering, University of Zagreb, Pierottijeva 6, 10000 Zagreb, Croatia; ivo.galic@rgn.hr (I.G.); ivandragicevic007@gmail.com (I.D.)

[2]  HGE-Solutions j.d.o.o. Radnička 2c, 10315 Novoselec, Croatia; miso@hges.hr

\*  Correspondence: ivica.pavicic@rgn.hr; Tel.: +38-59-9805-1725

**Featured Application: The presented methodology is a relatively fast and low-cost method that gives solid input into the state of the investigated rock mass, bedding orientation, degree of jointing, and preliminary block size estimation. All these parameters are very important for decision-making in the initial phase of quarry investment since these factors control the potential of the location for dimension stone deposit and type of excavation. The methodology is also applicable to hydrogeological (fractured aquifers), geotechnical, civil engineering, and engineering geology research (rockfalls, construction of roads, viaducts, railways, bridges, tunnels, etc.) where knowledge about fracture systems in the rock mass is crucial for further works.**

**Abstract:** The successful exploration of dimension stone mainly depends on the quality, size, and shape of extractable blocks of dimension stone. The investigated area is in the Pelješac Peninsula (Croatia), in the External Dinarides orogeny, built from thick carbonate succession, characterized by relatively small deposits of high-quality dimension stone. These conditions demand challenging geological investigations in the "pre-quarry" phase to find optimal quarry location. The size and shape of dimension stone blocks are mainly controlled by fracture pattern systems. In the rugged, covered terrains, it is very hard to obtain a satisfactory amount of fracture data from the surface, so it is necessary to collect them from the underground. Borehole camera technology can visualize the inner part of the rock mass and measure the fracture characteristics. The main conclusions are as follows: (1) the digital borehole camera technology provides a quick, effective, and low-cost geological survey of fractured rock mass; (2) statistical fracture distribution parameters, $P_{10}$, fracture spacing, Volumetric Joint Count ($J_v$) based on borehole wall survey can reflect the integrity of rock mass, providing a solid decision-making base for further investment plans and dimension stone excavation method.

**Keywords:** borehole camera; fracture patterns; overturned anticline; dimension stone; External Dinarides

## 1. Introduction

Dimension stone and crushed rocks are considered key natural resources that enable the sustainable functioning of the worldwide economy. Global demand for dimension stone, due to its aesthetic and petrophysical properties being used in the construction industry, including indoor and outdoor spaces, is related to the development of the economy [1]. A global increase in demand for dimension stone increased research activities worldwide, and thus in the Dinarides in Croatia and Bosnia and Herzegovina. The Dinarides are a fold-and-thrust orogenic belt that is geotectonically positioned along the NE margin of the Adriatic Sea. The Dinarides have great potential for dimension stone of sedimentary origin, but locations and the lateral extent of the potential areas are primarily controlled by regional tectonic settings. The External Dinarides are represented by a thick succession

of platform carbonates, which are very potential for dimension stone, proven by a dozen smaller quarries along the Adriatic coast. Since Dinarides are orogeny created by strong compressional tectonic forces, the rocks are often very faulted, folded, and fractured.

The result of all these natural factors is that dimension stone deposits are usually of high quality but relatively small (on a global scale). These conditions put more challenges on geological research to define the optimal position for quarry opening. The block size distribution is a crucial parameter for the profitable production of dimension stone within the deposit. The distribution of blocks and their sizes and shapes are mainly controlled by three-dimensional discontinuity systems (i.e., bedding planes, faults, fracture systems) [2]. In the research of dimension stone deposits, the particular emphasis should be on the distribution of discontinuities and their characteristics because fracture patterns mainly control block size distribution, type (surface pit or underground exploration), and exploitation success. In the initial phase of research, for the potential quarry, discontinuities are rarely available in a satisfactory amount for the spatial distribution analysis and statistics, due to terrain morphology, vegetation, karst features on the surface, and so forth. Every serious "pre-quarry" investigation includes several drill holes with cores, but it is not always possible to obtain discontinuity orientation from drill hole cores. Although all these factors are unfavorable, it is crucial to define some fracture pattern parameters, for preliminary Volumetric Joint Count estimation, as a key factor for reserve estimation and later profitability of the surface or underground exploration investment. Challenged by this situation, we have developed the method of defining fracture orientation parameters from the borehole camera video logging technique. Recently, borehole video logging technology (borehole camera, optical and acoustic televiewer) is widely used in geological and hydrogeological and nuclear waste research [3–10], civil engineering, and geotechnical engineering [8,9,11–16]. Geological characteristics of the rock mass such as lithology, fracture orientation, and morphology, karst phenomena (caves), rock weathering, and so forth can be detected by acquiring high-quality drill hole wall images [11,13]. In this paper, statistical analysis of the fracture pattern data obtained by borehole camera technology was examined to estimate expected block size and fracture set orientations. This is a relatively low-cost method for obtaining reliable preliminary fracture orientation data. Data were analyzed, and fracture sets with spatial parameters were defined: fracture sets, bedding, fracture set orientations, $P_{10}$ linear fracture intensity, fracture spacing, and Volumetric Joint Count ($J_v$). The proposed approach is a relatively low-cost methodology for fracture pattern data collection and analysis to obtain key parameters in their areas with a low amount of surface data. This is especially important in the initial phase of dimension stone quarry research. The presented methodology is not applicable only in the dimension and crushed rock research but also in the hydrogeological (fractured aquifers), geotechnical, civil engineering, and engineering geology research (rockfalls, construction of roads, viaducts, railways, bridges, tunnels, etc.).

## 2. Geological Settings

The investigated area is located on the E side of the Pelješac Peninsula, which is part of the External Dinarides, an approximately 600 km long fold-and-thrust orogenic belt [17–24]. The External Dinarides are geotectonically positioned along the NE margin of the Adriatic Sea, which corresponds with the NE part of the Adria Microplate [18,20,23–25]. The Dinarides orogenic belt was uplifted during the Late Eocene to Oligocene, due to the Adria Microplate–European Plate collision [18,19]. Most of the sediments in the External Dinarides were deposited during Mesozoic on the Adriatic Carbonate Platform from lower Jurassic to Upper Cretaceous [21,22]. The total thickness of the sediments is estimated to a few kilometers (in places up to 8 km) [21]. During their geological evolution, Dinarides had a polyphase uplift, due to multiple compressional tectonic regimes that formed present tectonic settings of the area [17,20,21].

The wider research area in the Pelješac Peninsula is composed of three stratigraphical and depositional units: Cretaceous carbonates, Paleogene carbonate, and clastic sediments

and quaternary clastic sediments [26–31] (Figure 1). Cretaceous carbonates are the dominant geological unit in the Pelješac Peninsula, and they are represented by limestones and dolomites subdivided into four chronostratigraphic units from Hauterivian to Maastrichtian age (Figure 1). Since the research area is composed of carbonates of Coniacian to Maastrichtian ($K_2^{3-6}$), they will be described in more detail. Upper Cretaceous carbonates are represented by thick-layered fossiliferous limestones and late diagenetic dolomites. Limestones are very fossiliferous, with remains of rudists and other mollusks, Echinoderms, Bryozoans, Hydrozoas, and Foraminifera (Figure 2A–C,E). Lithofacies features indicate the deposition in the forereef environment with a substantial inflow of material from the nearby reefs [26–32]. On top of this lithofacies, there are thick series of vertical and lateral interchange of limestones and dolomites. Limestones are described as thick layers of mudstone and fenestral mudstones. The upper part of the unit is characterized by a higher dolomitization rate and gray to dark grey dolomites (Figure 2D,E). The interchange of limestones and late diagenetic dolomites results from irregular dolomitization, so there is a frequent lateral and vertical change in lithology. The youngest geological units in the area are foraminiferal limestones of the Eocene age. They are transgressive to the older deposits, usually Upper Cretaceous limestones [26–32] (Figure 1).

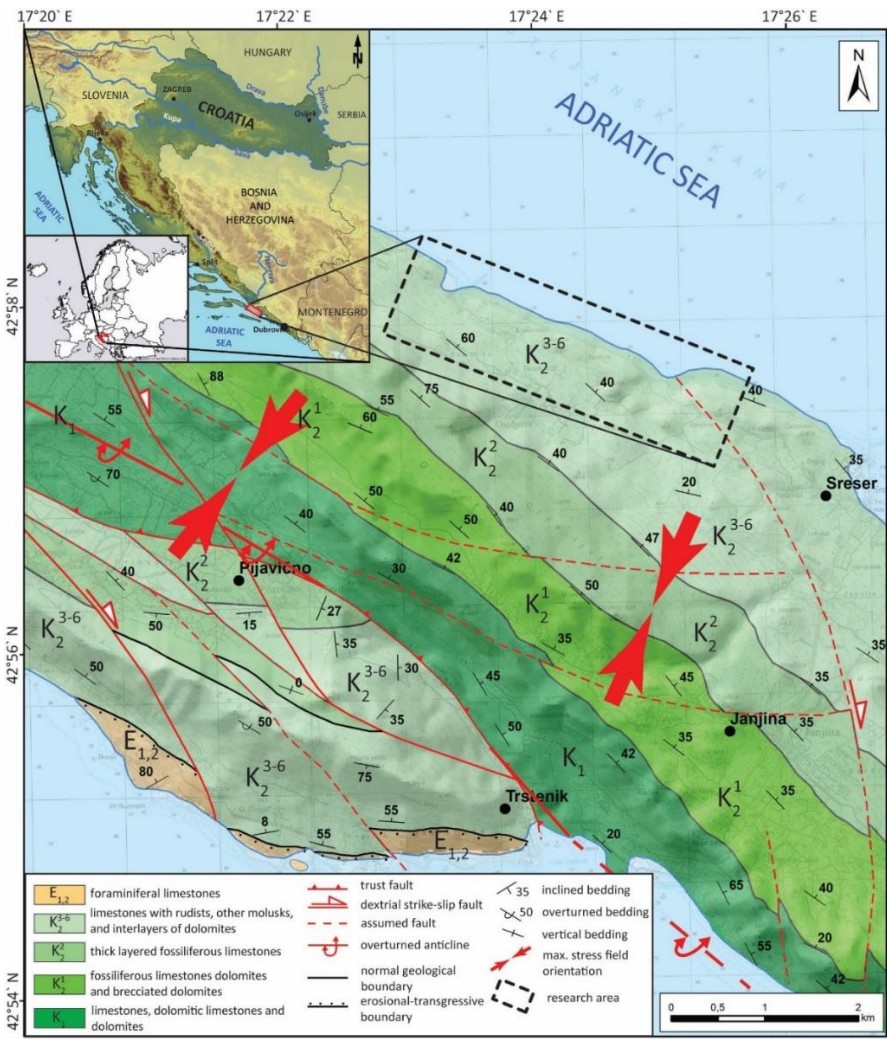

**Figure 1.** Geological map of the research area (modified after [26]). Stress orientation according to [20,27].

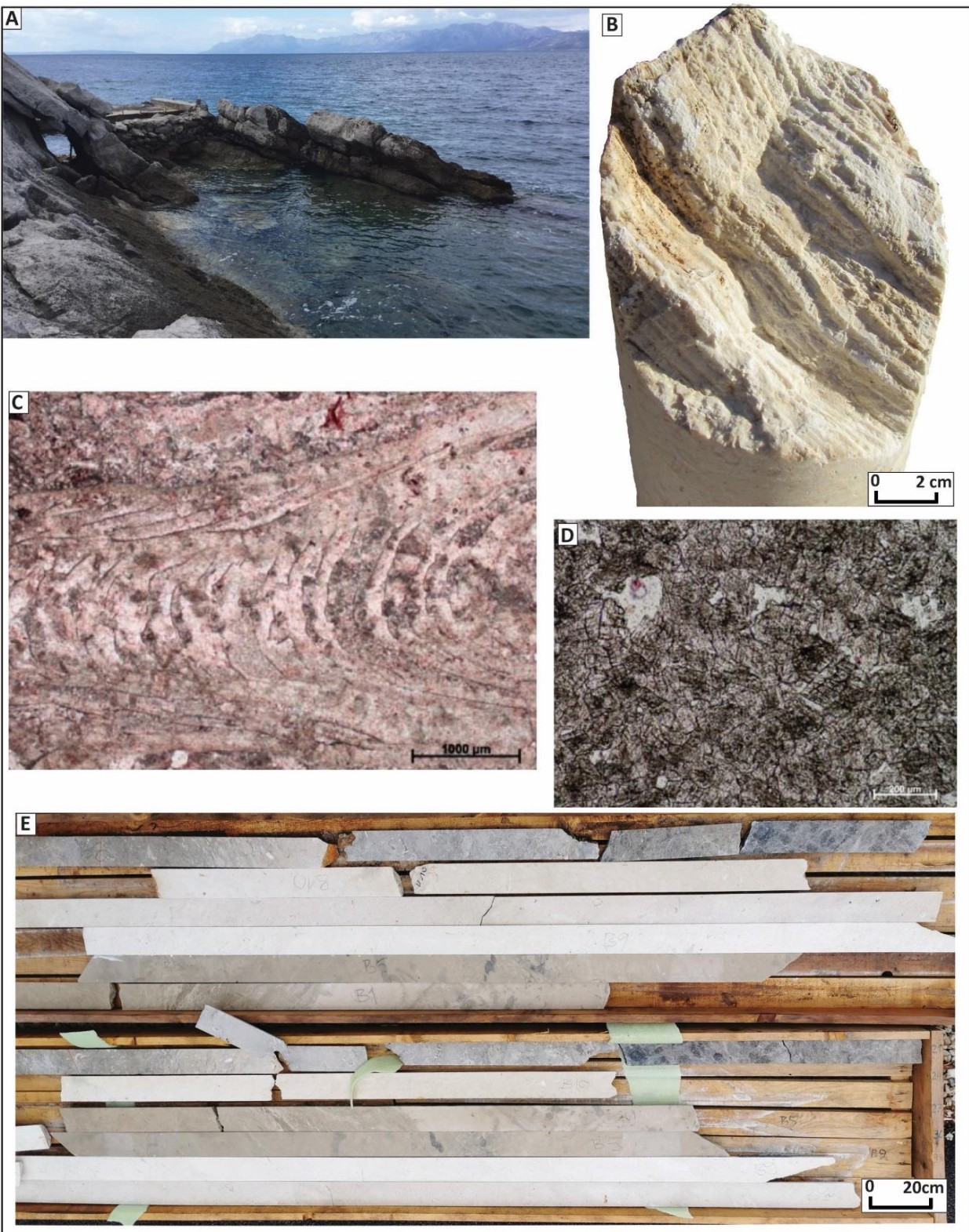

**Figure 2.** Shows (**A**) Upper Cretaceous limestones with 0.5–1 m thick layers in the northern part of Pelješac Peninsula; (**B**) Mollusk fossil in the drillcore; (**C**) microphotograph of fossiliferous limestone lithofacies; (**D**) microphotograph of late diagenetic dolomite lithofacies; (**E**) polished drill hole core samples of different lithofacies: white samples are micrite limestones, light gray to gray are fossiliferous limestones with bioturbations; dark samples are late diagenetic dolomites.

The Pelješac Peninsula is a relatively complex tectonic unit characterized by trusts, folds, and faults. Generally, it represents regional asymmetrical overturned anticline striking NW-SE. The core of the anticline contains the oldest, Lower Cretaceous deposits. The limbs are then composed of a succession of Lower Cretaceous to Paleogene sediments. Trusts are usually NW-SE striking with an inclination towards NE. These trusts are intersected by younger diagonal and transverse strike-slip faults. Our research area is placed in the SE limb of a regional anticline with NW-SE striking perpendicular to the maximal stress field (Figure 1).

## 3. Materials and Methods

The fracture orientation data can be collected in different ways: from the drill hole cores, the optical and acoustic televiewer survey, drill hole geophysics, and drill hole camera surveying. The best way to observe and measure fracture system parameters (orientation, density intensity, cross-cutting relations, etc.) is in the open pits, outcrops, and road cuts. When it is impossible to obtain the fracture data from the terrain, the only alternative option is to obtain them from the underground by drilling. Since terrain in our research area was impassable and had vegetation cover, and other drill hole analyses (geophysical, optical, and acoustic televiewer logging methods) are relatively expensive in the initial, "pre-quarry" phase of the research, improvisation was needed to find a relatively affordable way to measure the discontinuities orientation in the drill holes. In this research, ten drill holes were drilled, and drill hole cores were geologically logged (Figures 2 and 3). Drillhole cores were not undisturbed, which means that cores were rotated during drilling, so their orientation in space was not defined.

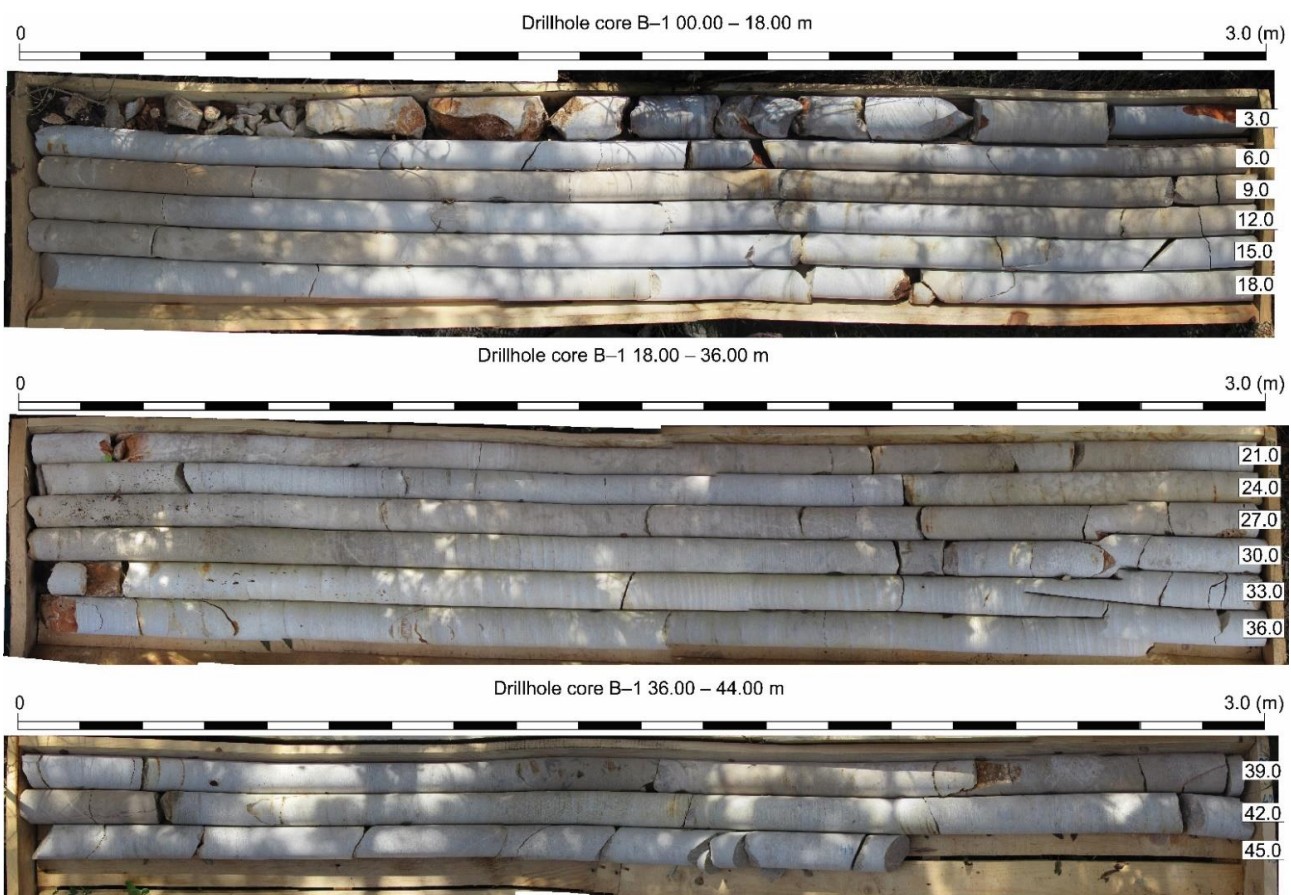

**Figure 3.** Example of drill hole core B-1.

### 3.1. Drillhole Camera Video Fracture Surveying

On the basis of the drill hole position and the drill hole core logs, six drill holes were chosen for the drill hole camera survey. We used a drill hole camera system with spatial orientation of the bottom and side camera (Figures 4 and 5). The borehole camera was transported by terrain vehicle to the drill hole location. The camera was then centered above the drill hole by a tripod, and the depth was set to zero m. Camera orientation was checked (on each drill hole) while the camera was still on the surface. After all initial checking was set up, the camera was lowered into the drill hole with a winch. Since the camera has a smaller radius than the drill hole, centering the camera was done with a tripod placed precisely above the drill hole. Camera stillness was partially established by the tripod and partially by a slow lowering of the camera to the drill hole. The camera needs to be still for acquiring precise orientation, so sometimes it was necessary to wait for the camera to stabilize. Alternatively, it is possible to use stabilizers between the camera and the drill hole wall. Still, it is dangerous in the open hole, but it can be dangerous due to the possible collapse of the drill hole wall (in the faulted or karstified intervals).

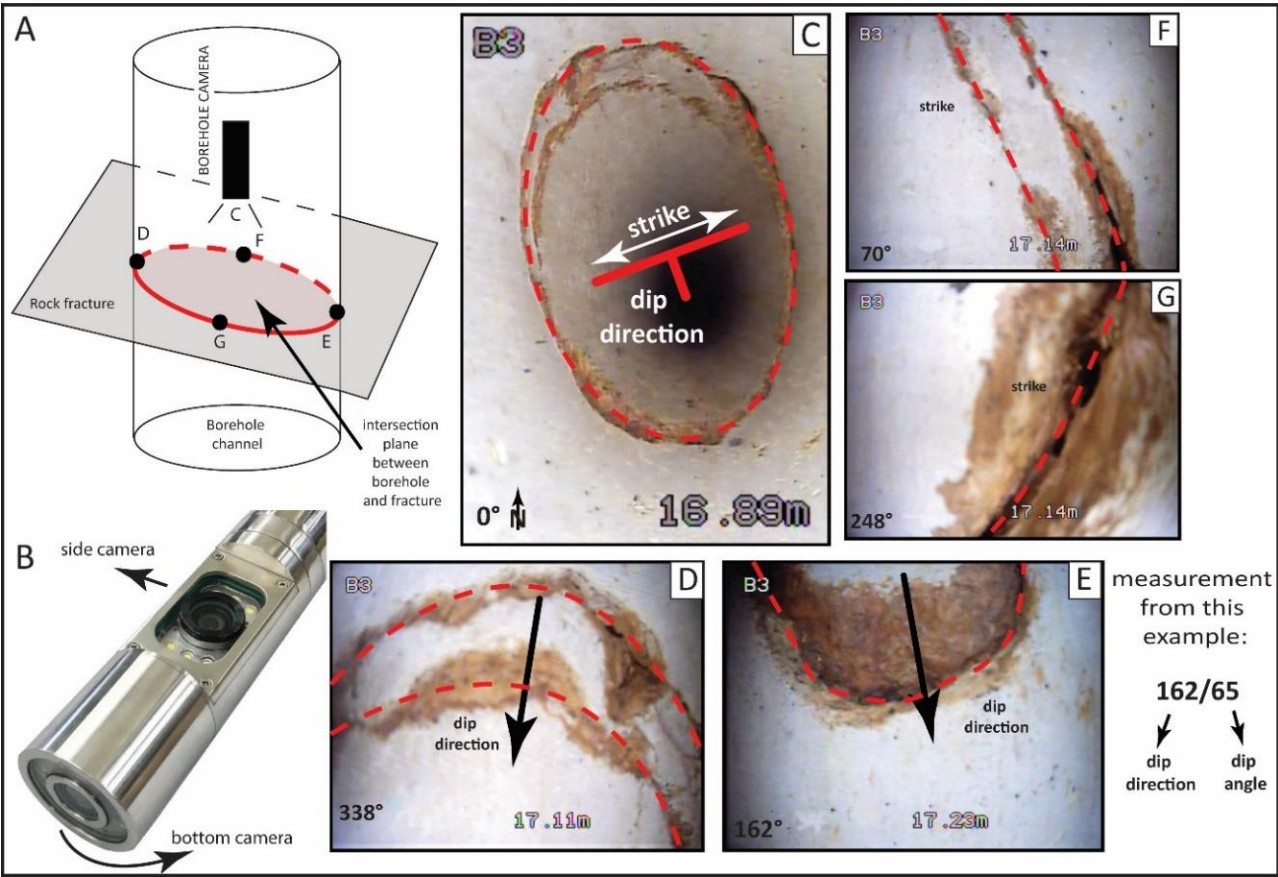

**Figure 4.** (**A**) Methodology of fracture orientation calculation. (**B**) Down camera view; (**C–G**) side camera view and fracture orientation determination.

The fracture surface is a three-dimensional plane in space with two main characteristics: dip direction and dip angle. Each fracture is measured manually in four points (D–G, in Figure 4A) by down and side cameras (Figure 4B), so fracture can be adequately defined in space. To define the orientation of the fracture, it would be enough to measure in three points. Still, because of measurement conditions (camera rotation, the angle between camera and fracture, subjectivity due to geologist experience), we take measurements in four points to better define the plane. Each fracture measurement was entered into a spreadsheet in the notebook. Measured dip angles were also checked on the drill hole

cores for higher precision of the measurement (Figure 3). First, from the bottom camera, we defined the strike and azimuth of the fracture, which is then checked and improved by the side camera (Figure 4A,B,D,E–G). The whole surveying was recorded so later interpretation, reinterpretation, and validation could be made. The whole process is repeated for each drill hole.

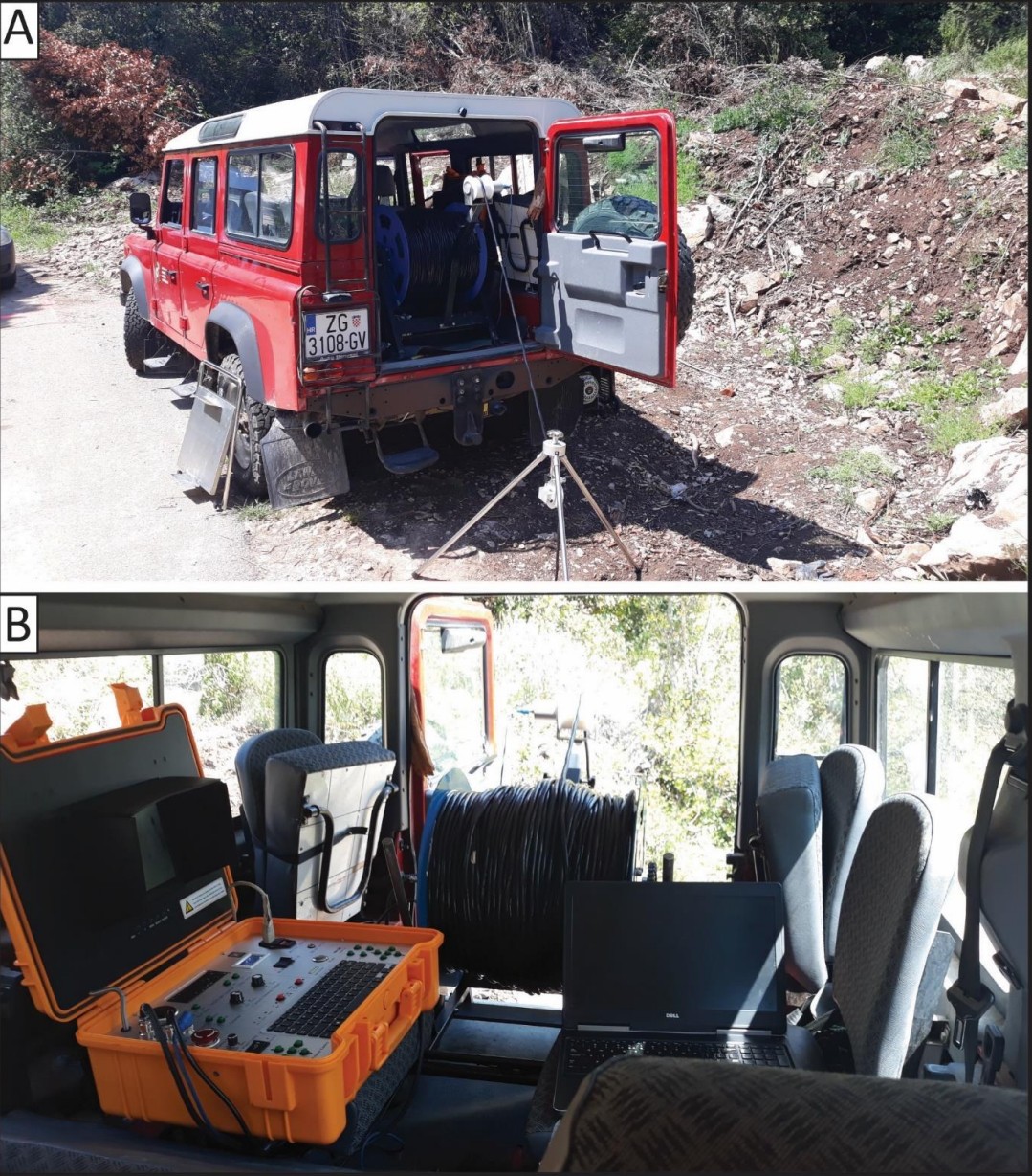

**Figure 5.** Borehole camera fieldwork configuration. (**A**) Terrain vehicle and tripod for centering of borehole camera; (**B**) operation console and winch for borehole camera.

*3.2. Statistical Fracture Distribution Parameters and Relation to the Associated Geological Structure*

The next steps were a stereonet-based structural analysis of the fracture data and grouping of the fractures into fracture sets. Fracture sets were defined by the orientation

and the Fisher coefficient of dispersion (k). Input data for calculating **k** are dip direction and dip of the discontinuities. Fisher coefficient was calculated by the Equation (1) [33]:

$$k = (n - 1)/(n - R) \tag{1}$$

where k is Fisher coefficient of dispersion, and n is the number of measurements, and R is the mean vector direction.

Fracture orientation, cluster analysis, and Terzaghi bias correction [34] were made with an academic license of Petroleum Experts Move software. Cluster analysis or clustering is the classification of objects into subsets (clusters) so that objects in each subset share the same properties. The used clustering technique is based on the k-means algorithm. The k-means algorithm is an algorithm to cluster objects-based attributes into k populations. To subdivide data into clusters, it was necessary to manually estimate the number of clusters based on stereo net observation. It is important to notice that this part is highly dependent on the interpreter's experience. This algorithm is relatively fast, so a common procedure is to run the algorithm several times and return the best clustering. The algorithm then separates the data into the number of expected clusters. Since there are not enough data from the surface to validate fracture sets defined by drill hole camera logging, we needed to validate our interpretation based on theoretical fracture models expected in the interpreted geological structure (example in [35,36]). When it is hard or impossible to have a statistically meaningful amount of the fracture measurements, it is necessary to obtain as much data about the geological structure since fractures and fracture patterns result from the geological settings of the area [37]. Systematic fractures usually show close relationships to faults and folds, which develop during the same tectonic event [35]. On the basis of the structure interpretation, we can anticipate the fracture patterns related to the position of our research area in the structure (Figure 6).

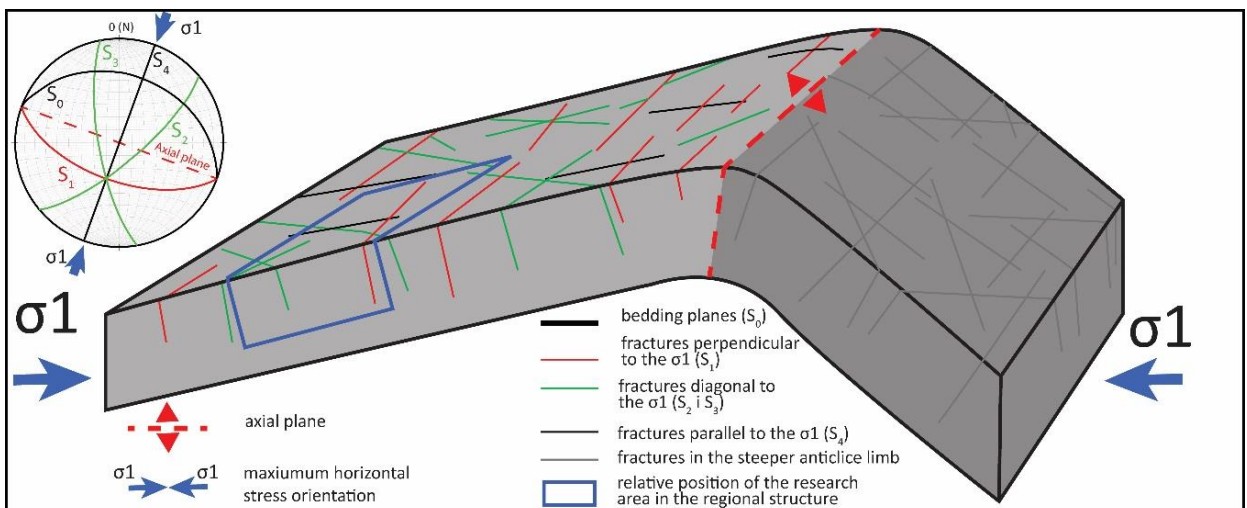

**Figure 6.** Predicted fracture systems on an anticline. Expected fracture orientations are based on models in [35,36].

Fracture system parameters, like density and intensity, depending on the lithology, layer thickness, mechanical characteristics of rocks, structural position, fold curvature, distance from faults, and so forth [38–41]. For the preliminary rock mass evaluation, the following parameters are calculated:

- $P_{10}$—number of fractures per unit distance (in our example 1 m drill hole length) ($m^{-1}$) (Figure 7)

–  Apparent ($S_m$) and true fracture spacing (d) in each individual set (m) (Figure 7)—if fracture orientation parameters and apparent spacing between fracture in the set are known, we can calculate true spacing by equation:

$$d = S_m \times \sin\delta \tag{2}$$

where d is true fracture spacing, $S_m$ is apparent fracture spacing, and $\delta$ is an angle between fracture plane and drill hole axis.

–  Volumetric Joint Count ($J_v$)—number of fractures in the unit volume of the rock mass ($m^{-3}$) [29–34]. This parameter can be calculated from the true fracture spacing [42]. Volumetric Joint Count represents a measure of fracture density in the rock volume (in the literature, it can also be marked as $P_{30}$):

$$J_v = 1/d_1 + 1/d_2 + \ldots + 1/d_n \tag{3}$$

where $J_v$ is Volumetric Joint Count ($m^{-3}$), and d is the average spacing between fractures in the set (m).

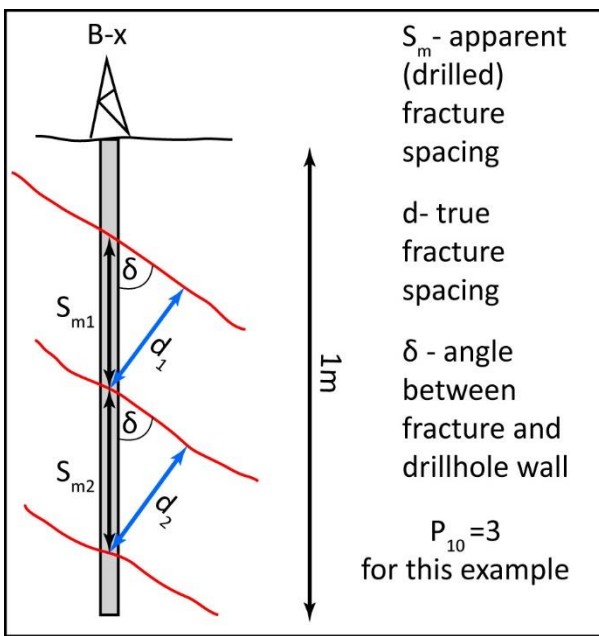

**Figure 7.** The relation between apparent (drilled) and true fracture spacing.

## 4. Results

### 4.1. The Analysis of Fracture Orientation

For this research, six drill holes were surveyed by a video camera with a cumulative depth of 378 m. By this method, 246 discontinuities were measured in the drill holes. Even though the terrain was almost inaccessible, we have done discontinuity mapping, and 48 discontinuities on the surface were measured. It is important to note that most of the surface data are bedding measurements and very few fracture measurements on the surface (only nine measurements of $S_1$ fractures, four measurements of $S_2$ fractures, and one of the $S_3$ fractures) due to vegetation coverage karstification and inaccessibility of the surface. Both surface and subsurface surveys resulted in a total of 294 discontinuity measurements.

On the basis of the cluster analysis of fracture orientation data, we classified them into four sets (Table 1, Figure 8), from which $S_0$ is the bedding and $S_1$, $S_2$, and $S_3$ are fractures systems.

**Table 1.** Fracture orientation data and fracture set interpretation.

| Set | | Drillhole Data | | | | |
|---|---|---|---|---|---|---|
| | | Dip Direction | Dip Angle | No. of Measurements | Standard Deviation (°) | Fisher |
| Bedding | $S_0$ | 15 | 46 | 16 | 55.99 | 41.18 |
| Fractures | $S_1$ | 228 | 73 | 58 | 53.89 | 16.89 |
| | $S_2$ | 86 | 72 | 104 | 53.95 | 18.08 |
| | $S_3$ | 319 | 76 | 35 | 53.61 | 15.60 |
| Set | | Terrain Data | | | | |
| | | Dip Direction | Dip Angle | No. of Measurements | | Fisher |
| Bedding | $S_0$ | 29 | 36 | 33 | 55.47 | 30.41 |
| Fractures | $S_1$ | 227 | 61 | 9 | 54.52 | 12.82 |
| | $S_2$ | 114 | 72 | 4 | 52.25 | 8.52 |
| | $S_3$ | 0 | 60 | 1 | - | - |
| Set | | All Measurements | | | | |
| | | Dip Direction | Dip Angle | No. of Measurements | | Fisher |
| Bedding | $S_0$ | 20 | 39 | 49 | 55.40 | 29.65 |
| Fractures | $S_1$ | 226 | 71 | 67 | 53.84 | 16.34 |
| | $S_2$ | 93 | 70 | 108 | 53.77 | 16.09 |
| | $S_3$ | 319 | 76 | 36 | 53.61 | 15.05 |

Fracture system orientations and their other spatial distribution parameters are strongly dependent on the position in the geological structure (regional and local) [35,36,43,44]. Expected fracture orientations in different parts of the folds are well defined in the scientific literature [35]. Nonetheless, it is important to note that every research area was subdued to different geological processes with various intensities during its evolution, so not all theoretical fracture systems are necessarily formed. Fracture system orientation in the fold structures is controlled by many factors: scale, fold type, layer thickness, mechanical properties of rocks, lithology, stress orientation, and so forth [35,44]. Theoretically, in the gentle anticline limb, we can expect two tension fracture sets and two shear fracture sets [35]. It is important to note that orientation and relationship between fracture sets, as well as a number of developed sets, are controlled by position in the anticline structure (gentle limb, hinge, or steeper limb) [7,35]. Our research area is in the gentle limb of an overturned anticline where bedding is generally inclined towards NE by 40° (Figures 1, 2, 6 and 7; Table 1). Maximal compressional stress orientation is generally NE-SW (Figure 1) [20,27,45]. In this structure, the $S_1$ fracture set corresponds to the tension fracture set with the same strike as the axial plane and bedding (Figures 6 and 8). Fracture sets $S_2$ and $S_3$ correspond to the shear fracture conjugate sets with an angle between them 65°, and they are diagonal to the $\sigma_1$. The angle between $S_2$ and $\sigma_1$ is 32° and between $S_3$ and $\sigma_1$ is 33.2°. Our drill hole video survey discovered that all fracture sets have a visible fracture aperture with occasional clay infill, which indicates that these fractures originate from tension stress. We interpreted this phenomenon as tension developed during the exhumation of these rocks to the surface during formation of the Dinarides. On the basis of these results, there is a minimum of three fracture sets and bedding in our research area that can be correlated to the position in the gentle limb of an anticline. Fracture set $S_4$ in the Price, 1966 theoretical model (Figure 7) perpendicular to the axial plane is not developed in our case.

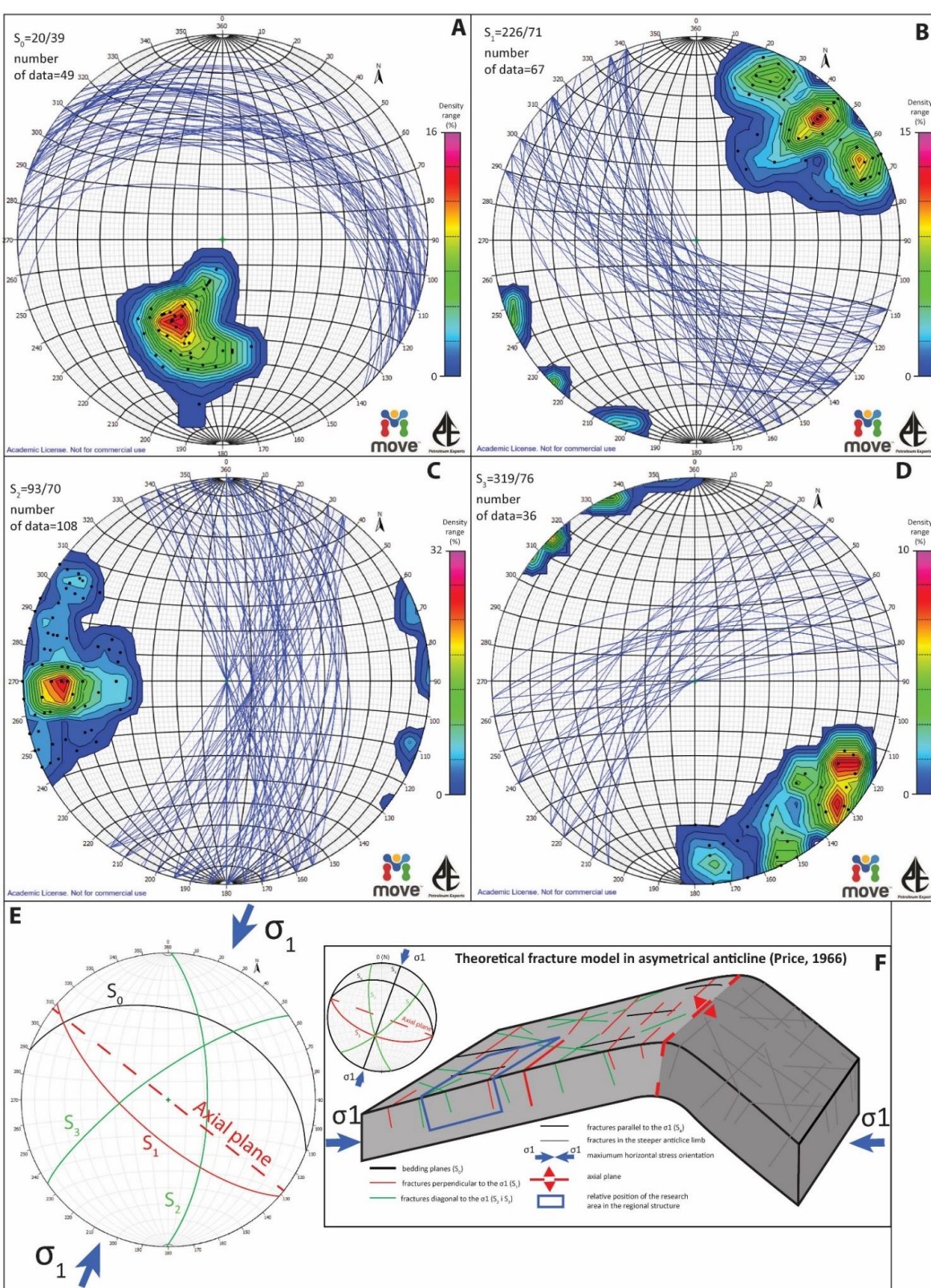

**Figure 8.** Stereonets of measured discontinuities: (**A**) bedding; (**B**) $S_1$ fracture set; (**C**) $S_2$, fracture set; (**D**) $S_3$ fracture set; (**E**) stereonet with an average orientation of each set and axial plane; (**F**) theoretical fracture system in asymmetric anticline according to [35,46].

### 4.2. Statistical Analysis of Spatial Distribution Parameters of Fractures

4.2.1. Linear Density/Intensity of Fractures—$P_{10}$

The linear density/intensity of fractures ($P_{10}$) represents a number of fractures per unit of length (in our case, drill hole axis). After we defined preliminary fracture sets, it was possible to calculate $P_{10}$ for each set in the drill hole and each set overall (Table 2). The highest average $P_{10}$ was calculated for the $S_2$ fracture set with value 0.39 m$^{-1}$ and lowest for $S_3$ with value 0.11 m$^{-1}$.

**Table 2.** Linear density/intensity of fractures—$P_{10}$.

| Drillhole | Fracture Set | Length of the Interval (m) | Number of Fractures Per Interval | $P_{10}$ m$^{-1}$ |
|---|---|---|---|---|
| B-1 | $S_1$ | 33 | 5 | 0.15 |
| B-3 | $S_1$ | 79.4 | 6 | 0.08 |
| B-5 | $S_1$ | 44.5 | 11 | 0.25 |
| B-6 | $S_1$ | 24.5 | 18 | 0.73 |
| B-7 | $S_1$ | 67 | 16 | 0.24 |
| B-9 | $S_1$ | 67 | 13 | 0.19 |
| B-1 | $S_2$ | 33 | 13 | 0.39 |
| B-3 | $S_2$ | 65.3 | 15 | 0.23 |
| B-5 | $S_2$ | 25 | 7 | 0.28 |
| B-6 | $S_2$ | 27.5 | 12 | 0.44 |
| B-7 | $S_2$ | 44.7 | 22 | 0.49 |
| B-9 | $S_2$ | 73.8 | 38 | 0.51 |
| B-1 | $S_3$ | 28.1 | 0 | 0 |
| B-3 | $S_3$ | 56 | 9 | 0.16 |
| B-5 | $S_3$ | 44.5 | 0 | 0 |
| B-6 | $S_3$ | 20 | 4 | 0.20 |
| B-7 | $S_3$ | 56 | 17 | 0.30 |
| B-9 | $S_3$ | 73.8 | 0 | 0 |
| Average $P_{10}$ per set | | | | |
| Set | | | $P_{10}$ | |
| $S_1$ | | | 0.27 | |
| $S_2$ | | | 0.39 | |
| $S_3$ | | | 0.11 | |

However, it is important to note that this parameter quantifies the only apparent number of fractures per unit of length since the distance between two fractures can be apparent if the drill hole channel is not perpendicular to the fracture orientation (Figure 7). By this parameter, it is not possible to calculate real spacing between fractures.

4.2.2. Apparent and True Fracture Spacing

Apparent fracture spacing ($S_m$) for all sets is in the interval between 0.5 and 4.39 (Table 3). Thus, fracture spacing is not regular in space, visible in the $S_m$ calculated for each set in each drill hole (Table 3). Average values of $S_m$ for $S_1$ set is 5.67 m, for $S_2$ from 3.24 m, and for $S_3$ 5.14 m. The average value of $S_m$ for bedding ($S_0$) is 5.54. Average true fracture spacing (d) for $S_1$ is 2.06 m; for $S_2$, parameter d is 1.24 m, and for $S_3$ from 1.32 m. The bedding have the highest values of true spacing, 3.74 m, which is consistent with our fieldwork data and literature descriptions of these rocks in other localities, where these carbonates are described as thick layered [26,28–32].

**Table 3.** Spatial distribution properties: apparent fracture spacing ($S_m$), average angle between fracture and drill hole axis and true fracture spacing (d) for each interpreted discontinuity set, and Volumetric Joint Count ($J_v$) for the discontinuity pattern system.

| Drillho | Fracture Set | Drillhole Depth/Interval Length | Apparent Fracture Spacing | The Average Angle between Fracture and Drill Hole Axis | Sinδ | True Fracture Spacing | 1/d |
|---|---|---|---|---|---|---|---|
| | | (m) | $S_m$ (m) | ° | | d (m) | $m^{-1}$ |
| B-3 | $S_0$ | 79.40 | 4.20 | 45.00 | 0.71 | 2.97 | 0.34 |
| B-6 | $S_0$ | 53.50 | 2.90 | 40.00 | 0.64 | 1.86 | 0.54 |
| B-7 | $S_0$ | 83.00 | 7.70 | 40.00 | 0.64 | 4.95 | 0.20 |
| B-9 | $S_0$ | 73.80 | 7.37 | 45.00 | 0.64 | 5.21 | 0.45 |
| B-1 | $S_1$ | 44.00 | 8.42 | 31.41 | 0.52 | 4.39 | 0.23 |
| B-3 | $S_1$ | 79.40 | 11.92 | 17.70 | 0.30 | 3.63 | 0.28 |
| B-5 | $S_1$ | 44.50 | 2.93 | 12.80 | 0.22 | 0.65 | 1.54 |
| B-6 | $S_1$ | 53.50 | 1.73 | 16.61 | 0.29 | 0.50 | 2.02 |
| B-7 | $S_1$ | 83.00 | 3.93 | 17.67 | 0.30 | 1.19 | 0.84 |
| B-9 | $S_1$ | 73.80 | 5.08 | 23.63 | 0.40 | 2.04 | 0.49 |
| B-1 | $S_2$ | 44.00 | 3.15 | 29.88 | 0.50 | 1.57 | 0.64 |
| B-3 | $S_2$ | 79.40 | 4.82 | 11.81 | 0.20 | 0.99 | 1.01 |
| B-5 | $S_2$ | 44.50 | 2.76 | 25.83 | 0.64 | 1.77 | 0.56 |
| B-6 | $S_2$ | 53.50 | 3.89 | 26.91 | 0.45 | 1.76 | 0.57 |
| B-7 | $S_2$ | 83.00 | 2.72 | 10.40 | 0.18 | 0.49 | 2.03 |
| B-9 | $S_2$ | 73.80 | 2.13 | 23.71 | 0.40 | 0.86 | 1.17 |
| B-3 | $S_3$ | 44.00 | 6.70 | 10.25 | 0.18 | 1.19 | 0.84 |
| B-6 | $S_3$ | 53.50 | 5.01 | 21.67 | 0.37 | 1.85 | 0.54 |
| B-7 | $S_3$ | 83.00 | 3.72 | 14.47 | 0.25 | 0.93 | 1.08 |
| | Bedding $S_0$ | | 5.54 | 42.50 | 0.68 | 3.74 | 0.26 |
| | Fracture set $S_1$ | | 5.67 | 19.97 | 0.34 | 2.06 | 0.90 |
| | Fracture set $S_2$ | | 3.24 | 21.42 | 0.40 | 1.24 | 1.00 |
| | Fracture set $S_3$ | | 5.14 | 15.46 | 0.27 | 1.32 | 0.82 |
| | Volumetric Joint Count ($m^{-3}$) | | | | | 2.97 | |

### 4.2.3. Volumetric Joint Count ($J_v$)

After average values for true fracture spacing were defined, $J_v$ was easy to calculate by Equation (3). Ultimately, the Volumetric Joint Count for our example of four discontinuity sets (bedding and three fracture sets) is 2.97 fractures per $m^3$ (Table 3).

The rock mass can be classified based on $J_v$, from a very low degree of jointing to crushed rock mass (Table 4) [47]. Our example belongs to a small degree of jointing, which means that the area is definitely potential for dimension stone deposit.

**Table 4.** Classification of degree of joint of rock mass according to the Volumetric Joint Count (Jv) [47].

| Class | Very small | Small | Moderate | Large | Very Large | Crushed |
|---|---|---|---|---|---|---|
| $J_v$ ($m^{-3}$) | <1 | 1–3 | 3–10 | 10–30 | 30–60 | >60 |

### 5. Discussion

The productivity of dimension stone quarry mainly depends on the extractable block size and shape [2]. The shape and volume of the blocks are crucial factors and are controlled by the discontinuity pattern [2]. The External Dinarides orogeny is mostly built from a

thick succession of often tectonically disturbed carbonates, which resulted in relatively small deposits (on the global scale) of high-quality dimension stone. These conditions demand challenging geological investigations in the "pre-quarry" phase to find the optimal quarry location. All necessary surface geological investigations were made (geological mapping, stratigraphy, structural research, petrography, mechanical properties), but since the terrain is rocky, impassable, and covered by vegetation, a small number of fractures was measured. Drillhole geophysics and the Optical and Acoustic Televiewer surveys were too expensive, so we were challenged to find a new method to measure fractures in the drill holes. A borehole camera with the bottom and side camera's spatial orientation was used to measure fracture orientation parameters (Figures 4 and 5). Fractures were measured, then analyzed in stereonets by cluster analysis to define fracture sets. Four preliminary sets were defined, of which one is bedding ($S_0$), one is tensional fracture set ($S_1$) perpendicular to the stress orientation and subparallel to the fold axis, and two sets were defined as conjugated shear fracture sets ($S_2$ and $S_3$). It is important to notice that all sets have visible apertures at all depths, sometimes larger than eight cm and with red clay infill. We interpreted this as a result of the tension created by rock exhumation during the formation of the Dinarides. This can also be a reason for small offsets in parallelism between bedding, fold axis, and $S_2$ tension fractures. Thus, this error could result from the measurement error, which could not be confirmed. Fisher coefficients of dispersion are highest for the bedding, around 30, which indicates that bedding has the smallest dispersion of data around the mean value. This was expected, since most bedding measurements were taken from the surface (Table 1). Relatively low values of the Fisher coefficient, in the range between 15 and 20, for other fracture sets indicate relatively high dispersion around mean value. Since we had no way to validate the measurements, we tried to tie our fracture sets to the position in the geological structure [35–37]. Our research area is part of the gentle limb of a regional overturned asymmetric anticline, so we correlated the theoretical fracture model [35] for the asymmetric anticline to our results (Figure 8), and it generally fits the model. Low values of the Fisher coefficient of dispersion can result from multiple factors: local undulation of fractures, measuring error, the existence of more fracture sets than we interpreted, and so forth. This can be further investigated by the analysis of the fracture system on each drill hole in each lithofacies.

By calculating spatial distribution parameters, $P_{10}$, apparent spacing, true spacing, and $J_v$, we wanted to make a bold prediction of fracture spacing and block size in the potential deposit. These results are directly dependent on the measures and fracture set interpretation. Volumetric Joint Count for this case study is 2.97 fractures per $m^3$. By importing this value to the Palmstrom $J_v$-Block size correlation diagram [47], the estimated average block size in the deposit is about 1.5 $m^3$, which we characterize as potential in given geological conditions (bedding inclination of 40°, position in the orogeny).

The presented methodology is a relatively fast and low-cost method that gives solid input into the state of the investigated rock mass, bedding orientation, degree of jointing, fracture infill, Volumetric Joint Count, and preliminary block size estimation. All these parameters are very important for decision-making in the initial phase of quarry investment, since these factors control the potential of the location for dimension stone deposit and type of excavation. The methodology is applicable in the areas where deposits are relatively small, so large initial investments are not expected. The advantages of the method, compared to similar techniques, are price, speed, and no need for heavy machinery. The main disadvantages are precision (compared to the Optical or Acoustic Televiewer), and the results strongly depend on the interpreter's experience. The methodology is also applicable to hydrogeological (fractured aquifers), geotechnical, civil engineering, and engineering geology research (rockfalls, construction of roads, viaducts, railways, bridges, tunnels, etc.) where knowledge about fracture systems in the rock mass is crucial for further works.

We aimed to define the preliminary fracture pattern system and give a bold estimation of average block size, on the basis of which potential investor can decide to continue or abort the further research and decide which kind of exploitation technique can be

expected (open pit or underground excavation). It is important to note that these results can be characterized as preliminary and need to be updated with all new data in future research (open pits, galleries, cuts, etc.) to make a better fracture interpretation that will minimize financial losses during exploration. The results of this research have confirmed the justification of the previous research and supports further investments.

## 6. Conclusions

In the research area in Pelješac Peninsula (Croatia), intensive geological research was conducted to define the potential of the location for dimension stone quarry. Special emphasis was put on defining fracture pattern parameters, since fractures are crucial factors that control block size and shape. The terrain was very karstified and covered by vegetation, so it was impossible to obtain a meaningful amount of fracture analysis data. Challenged by the given task, a new methodology was proposed to survey fractures on drill hole walls by the borehole video camera and structural analysis of fracture sets that result in Volumetric Joint Count ($J_v$) estimation. General conclusions are:

(1) Borehole camera technology is a relatively low-cost method compared with other drill hole surveying techniques for acquiring fracture orientation data. Although the measured data quality is lower than the Optical or Acoustic Televiewer, with drill hole core data, fracture measurements on the terrain, and precise interpretation of geological structure, these methods give a satisfactory basement for preliminary fracture pattern interpretation.

(2) Besides fracture orientation, by borehole camera, it is possible to extract the rock mass condition, fracture aperture, fracture infill, cross-cut relations between fractures, and karst forms (i.e., caves and caverns), fracture surface roughness estimation, lithological features of rocks, and so forth.

(3) We preliminary define four discontinuity sets (bedding and three fracture sets) that correspond with the position in the gentle limb of an overturned anticline, which partially confirms that the survey was successful. After fracture orientation measurements, the following parameters of spatial distribution were calculated: Linear fracture intensity ($P_{10}$), Apparent Fracture Spacing ($S_m$), True Fracture Spacing (d), and Volumetric Joint Count ($J_v$), which indicate a small degree of jointing that is convenient for dimension stone deposit.

A fracture pattern system's complexity can generate various problems concerning opening a quarry, type of excavation method, and defining the ongoing mining [2,47]. Dinarides are usually characterized by small deposits of high-quality dimension stone. With that in mind, large investments in the extensive geological research in the underground research (optical and acoustic televiewer survey, drill hole geophysics) prior to the quarry's opening are not to be expected. The described method is a relatively fast and low-cost method that gives solid input into the state of the rock mass, bedding orientation, degree of jointing, and block size, which is very important for decision making, since it controls the locality's potential for dimension stone deposit and type of excavation.

**Author Contributions:** I.P. made the concept of the methodology and planned the fieldwork. I.P. and M.K. conducted fieldwork. I.G. and I.D., I.P. and I.G. contributed significantly to the manuscript preparation; M.K. contributed with figures and table preparation; I.D. significantly contributed to review and editing of the manuscript, I.G. contributed to the project administration and founding. All authors have read and agreed to the published version of the manuscript.

**Funding:** The APC was partially funded by I.G.

**Informed Consent Statement:** Informed consent was obtained from all subjects involved in the study.

**Data Availability Statement:** Data available on request due to privacy restrictions. The data presented in this study are available on request from the corresponding author.

**Acknowledgments:** The authors would like to thank company VICI-VENTUS d.o.o. for providing us with the data and permission to publish the results of this study. We are grateful to the Faculty of Mining, Geology and Petroleum Engineering, University of Zagreb, and the University of Zagreb for the partial funding of the publication. Furthermore, we are very grateful to the company HIDRO-GEO PROJEKT and the director of the company Perica Vukojević for lending us the drill hole camera. We are in great debt to the Petroleum Experts for donated academic licenses of Move software that allowed us to conduct a structural analysis of fracture orientations. We are grateful to reviewers for their constructive remarks and comments that significantly improved the quality of this paper.

**Conflicts of Interest:** The authors declare no conflict of interest.

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
