# Peer review of "Fracture System and Rock-Mass Characterization by Borehole Camera Surveying: Application in Dimension Stone Investigations in Geologically Complex Structures"

_applsci, doi:10.3390/app11020764_

Round 1

Reviewer 1 Report

Statement in line 258 and 259 is not correct.

Highest P10 was calculated for S2 fracture set with value 0.39? m-1 (0.51) and lowest for S3 with value 0,22? m-1(0.16). ?? check the values in Table 2.

Statement in line 283 and 284 need to be checked or recalculated again.

Volumetric Joint Count for our example, four discontinuity sets (bedding and three fracture sets) is 2,34 fractures per m3. ?? It is not clear how the value for Jv (m-3)??? was calculated in Table 4.

Author Response

Answers to Reviewer 1

Dear Reviewer,

Thank You very much for Your effort to revise and improve this manuscript with Your comments and remarks. Here are the answers:

Statement in line 258 and 259 is not correct. – acknowledged and corrected. Highest and lowest average P10

Highest P10 was calculated for S2 fracture set with value 0.39? m-1 (0.51) and lowest for S3 with value 0,22? m-1(0.16). ?? check the values in Table 2. – corrected, average P10 for entire set

Statement in line 283 and 284 need to be checked or recalculated again. - corrected

Answers to Reviewer 1

Dear Reviewer,

Thank You very much for Your effort to revise and improve this manuscript with Your comments and remarks. Here are the answers:

Statement in line 258 and 259 is not correct. – acknowledged and corrected. Highest and lowest average P10

Highest P10 was calculated for S2 fracture set with value 0.39? m-1 (0.51) and lowest for S3 with value 0,22? m-1(0.16). ?? check the values in Table 2. – corrected, average P10 for entire set

Statement in line 283 and 284 need to be checked or recalculated again. - corrected

Volumetric Joint Count for our example, four discontinuity sets (bedding and three fracture sets) is 2,34 fractures per m3. ?? It is not clear how the value for Jv (m-3)??? was calculated in Table 4. – corrected and all values are recalculated. It was calculated by the equation 2

Kind regards,

Authors

Volumetric Joint Count for our example, four discontinuity sets (bedding and three fracture sets) is 2,34 fractures per m3. ?? It is not clear how the value for Jv (m-3)??? was calculated in Table 4. – corrected and all values are recalculated. It was calculated by the equation 2

Kind regards,

Authors

Reviewer 2 Report

Nr.

Line

Comment

1

15

The factors controls --> remove s

2

15

controls the potential of the locality for dimension stone deposit

--> i suppose it is means, the “location” or local conditions?

3

16

The methodology

4

20-21

Maybe change how sentence is written – hard to understand.

5

87

... is located in the E margin of Pelješac Peninsula...

--> E margins means eastern side/area/part?

6

126-132

Is the source missing in this paragraph?

7

128

Sentence contains errors and is a bit hard to understand à Suggestion:
In the core of the anticline, Lower Cretaceous deposits are located. In the limbs, deposits vary…

8

144

… were detailly geologically logged in detail…

9

155

Why did you choose four points and not three points? (A am sure there is a practical reason, for the reader it would be good to understand this reason and workflow)

10

166

Sentence contains errors
Suggestion: The next steps were a stereo net based structural analysis of the fracture data and grouping of the fractures into fracture sets.

11

167

Fracture sets were defined by the orientation and the Fisher coefficient of dispersion. Fracture orientation and cluster analysis were conducted with academic license of Petroleum Experts Move software.

è Different cluster detection algorithms exist. Please explain more in detail which analysis was run.

è Also the formula for Fisher dispersion should be given as it is not totally clear what the input parameters are. P10, d, strike, dip?

12

191

Please use the proper formular writing function of your writing software (e.g. Word): e.g.

13

193

Where d is true fracture spacing, Sm is apparent fracture spacing and δ is angle between fracture plane and drillhole axis.

Also, this formula is only valid, when the drillholes are sufficiently vertical. Is this provided? It was not mentioned in the text.

14

201

same as for comment Nr. 12; also, please correctly use subscripting.

15

186

P10 – number of fractures per unit distance (in our example drillhole depth) (m-1) (Fig. 7)

è From your explanation, it is not totally clear how you defined P10. Is it number of fractures / 1 m drillhole length?

241

Fisher coefficients of dispersion are highest for the bedding, around 30, which indicate that bedding have the smallest dispersion of data around mean value.

è should a high coefficient of dispersion not result in a high dispersion?

158

…0,39 m-1

271

…4.82 m and…

286

Table 4.

è it is not clear how the final Jv is calculated. Following equation (2), I receive Jv of 2.86.

307

…stereonets…

General Remarks:

Generally, the paper presents an innovative and effective approach to utilize existing technologies to obtain valuable information in a slim workflow.

Especially the use of software requires the understanding of the mechanisms within the software. As such, the workflow within the Move software should be explained more in detail. Especially which kind of cluster analysis is used, what are the input parameters? The same should be done for the Fisher Measure of Dispersion.

From the context, the final results appear to be calculated as mean values. However, before that, the Fisher Dispersion as a measure of dispersion of the results was calculated. In the context of future mining prospects, the dispersion of the results is of high interest to assess the suitability of the new mining field. As such a measure of dispersion (e.g. standard deviation) that is widely used in the industry should be given for the final results as well. Additionally, geologic quality features are spatially distributed. While a spatial analysis of the results might be too much within the constraints of one paper, the results should at least be described/plotted with respect to their spatial relation. Then also in the outlook, it can be referred to future usage possibilities to analyse these data spatially in the resource estimation and mine planning context.

General correction needs:

Sub- and superscripts

The use of sub- and superscripts is not concise and should be corrected for the final submission.

Comma and Point

English uses point (.) as decimal separator. This needs to be corrected and consistent throughout the paper.

Articles

The use of the articles “the” and “a” should be revised to be consistent with the correct use of the English language. Minor errors in this regard can be found approximately in every second to third sentence.

Example:

Line 135 ff: The best way to observe and measure fracture system parameters (orientation, density intensity, cross-cutting relations etc.) are in the open pits, on the outcrops, road cuts or on a terrain with no soil or vegetation.

Author Response

Answers to Reviewer 2

Dear Reviewer,

Thank You very much for Your effort to revise and improve this manuscript with Your comments and remarks. Here are the answers:

Nr.

Line

Comment

1

15

The factors controls --> remove s - corrected

2

15

controls the potential of the locality for dimension stone deposit

--> i suppose it is means, the “location” or local conditions? – true, rephrased to location to be more understandable.

3

16

The methodology - corrected

4

20-21

Maybe change how sentence is written – hard to understand. – acknowledged and rephrased

5

87

... is located in the E margin of Pelješac Peninsula...

--> E margins means eastern side/area/part? - Exactly, rephrased to “E side”

6

126-132

Is the source missing in this paragraph? – reference added

7

128

Sentence contains errors and is a bit hard to understand à Suggestion:
In the core of the anticline, Lower Cretaceous deposits are located. In the limbs, deposits vary… - rephrased in “The core of the anticline contains the oldest, Lower Cretaceous deposits. The limbs are then composed from succession of Lower Cretaceous to Paleogene deposits. ..

8

144

… were detailly geologically logged in detail…corrected

9

155

Why did you choose four points and not three points? (A am sure there is a practical reason, for the reader it would be good to understand this reason and workflow) – of course, the only reason is to secure more precision, since the methodology itself already depends mostly on the interpreter. Text added ”To define the orientation of the plane it would be enough to measure in three points but because measurements condition (camera rotation, angle between camera and fracture, subjectivity due to geologist experience) we take measurements in four points to better define the plane. Each fracture measurement was entered into a spreadsheet in the notebook”

10

166

Sentence contains errors
Suggestion: The next steps were a stereo net based structural analysis of the fracture data and grouping of the fractures into fracture sets.- acknowledged and corrected and suggested by the reviewer. Thank You!

11

167

Fracture sets were defined by the orientation and the Fisher coefficient of dispersion. Fracture orientation and cluster analysis were conducted with academic license of Petroleum Experts Move software.

è Different cluster detection algorithms exist. Please explain more in detail which analysis was run.

Acknowledged and explained in the text, thank You! Text added:” Cluster analysis or clustering is the process of classification of objects into subsets (clusters) so that objects in each subset share same properties. Used clustering technique is based on k-means algorithm. The k-means algorithm is an algorithm to cluster objects-based attributes into k populations. To subdivide data into clusters it was necessary to manually estimate the number of clusters based on stereo net observation. It is important to notice that this part is highly dependent on the interpreter’s experience. This algorithm is relatively fast, so common procedure is to run the algorithm several times and return the best clustering found. The algorithm then separates the data into the number of expected clusters.

è Also the formula for Fisher dispersion should be given as it is not totally clear what the input parameters are. P10, d, strike, dip?

Acknowledged and explained in text, thank You! Text added:” The next steps were a stereo net based structural analysis of the fracture data and grouping of the fractures into fracture sets.Next step was to make structural analysis of fracture data in the stereo net and group fractures in fracture sets. Fracture sets were defined by the orientation and the Fisher coefficient of dispersion (k). Input data for calculating k are dip direction and dip of the discontinuities. Fisher coefficient was calculated by the equation [33] (1):

k = (n-1)/(n-R)                                                                           (1)

where k is Fisher coefficient of dispersion, and n is number of measurements and R is mean vector direction.

12

191

Please use the proper formular writing function of your writing software (e.g. Word): e.g. – We tried to use equation tool in the word template, but it is not working, and by their instructions this is the way to write equations. I guess it will be done during article processing, but I will ask the editor. I will also provide additional word document with all equations written in the equation tool!

13

193

Where is true fracture spacing, Sm is apparent fracture spacing and δ is angle between fracture plane and drillhole axis.

Also, this formula is only valid, when the drillholes are sufficiently vertical. Is this provided? It was not mentioned in the text. – True, since we were supervising the drilling and logging drillhole cores we did not notice any significant deviations. Same was concluded by camera inspection (there was no problems with camera jamming. But to be completely honest, it is not possible to know without deviation measurements which was of course not conducted so we assumed that the drillhole channel was vertical.

14

201

same as for comment Nr. 12; also, please correctly use subscripting. acknowledged

15

186

P10 – number of fractures per unit distance (in our example drillhole depth) (m-1) (Fig. 7)

è From your explanation, it is not totally clear how you defined P10. Is it number of fractures / 1 m drillhole length? – correct and improved as reviewer suggested

241

Fisher coefficients of dispersion are highest for the bedding, around 30, which indicate that bedding have the smallest dispersion of data around mean value.

è should a high coefficient of dispersion not result in a high dispersion? When Fisher coefficient k is applied high values mean uniform data and smallest is dispersion. Small k parameter means high dispersion and non-uniform. If the k is small (less than 10), data can hardly be classified in a one set.

158

…0,39 m-1 …corrected. Also, other sub and superscripts are corrected in the text and tables.

271

…4.82 m and…corrected

286

Table 4.

è it is not clear how the final Jv is calculated. Following equation (2), I receive Jv of 2.86. - corrected, after all recalculations of other parameters too, value is 2,97

307

…stereonets…corrected

General Remarks:

Generally, the paper presents an innovative and effective approach to utilize existing technologies to obtain valuable information in a slim workflow.

Especially the use of software requires the understanding of the mechanisms within the software. As such, the workflow within the Move software should be explained more in detail. Especially which kind of cluster analysis is used, what are the input parameters? The same should be done for the Fisher Measure of Dispersion. – done, text is updated by more detail description of the Move workflow.

From the context, the final results appear to be calculated as mean values. However, before that, the Fisher Dispersion as a measure of dispersion of the results was calculated. In the context of future mining prospects, the dispersion of the results is of high interest to assess the suitability of the new mining field. As such a measure of dispersion (e.g. standard deviation) that is widely used in the industry should be given for the final results as well. Additionally, geologic quality features are spatially distributed. While a spatial analysis of the results might be too much within the constraints of one paper, the results should at least be described/plotted with respect to their spatial relation. Then also in the outlook, it can be referred to future usage possibilities to analyse these data spatially in the resource estimation and mine planning context.

General correction needs:

Sub- and superscripts - corrected

The use of sub- and superscripts is not concise and should be corrected for the final submission. corrected

Comma and Point - corrected

English uses point (.) as decimal separator. This needs to be corrected and consistent throughout the paper. - corrected

Articles

The use of the articles “the” and “a” should be revised to be consistent with the correct use of the English language. Minor errors in this regard can be found approximately in every second to third sentence. – corrected by the native English speaker.

Example:

Line 135 ff: The best way to observe and measure fracture system parameters (orientation, density intensity, cross-cutting relations etc.) are in the open pits, on the outcrops, road cuts or on a terrain with no soil or vegetation.

Kind regards,

Authors

Reviewer 3 Report

The work presented is a low-cost methodology to measure discontinuities in boreholes. The work may be of interest, but to be published the manuscript must be considerably improved.

First, it is necessary to include in the introduction a more exhaustive bibliographic review of the methodology that is currently used, both commercially and academically, to measure discontinuities within surveys.

On the other hand, it is necessary to improve the methodology section. Mainly in the description of the method used. It is not stated where the cameras are located and what method is used to maintain the orientation of the cameras and how the orientation of the discontinuities is determined once the camera is rotated to take measurements.

Subsequently, in the discussion section, what is done is a summary of the results, however, an assessment of the proposed method should be made and the advantages over existing methods should be established.

In this work it is necessary to apply the Terzaghi Weighting (Terzaghi, 1965). Otherwise, the data obtained will have a bias because when orientation measurements are made, a bias is introduced in favor of those features which are perpendicular to the direction of surveying. Without this correction, the data obtained cannot be considered of quality.

Finally, there are a series of misprints and errors that need to be corrected:

  • A decimal commas is used throughout the text and must be replaced by a decimal point.
  • Line 39, Dimension stone cannot be considered as mineral resource.
  • Figure 1, graphic and numerical scale do not match.
  • Figure 1, In the legend there are no differences in the K1 lithologies
  • Figure 2A, no graphical scale is placed on a landscape photograph.
  • Lines 137-138, “terrain with no soil or vegetation” is the same as an outcrop.
  • Figure 6, what are Sd fractures?
  • Line 209, check english.
  • Table 1, there is data in this table that is not understood. For example, why are there only 16 So measures in the survey? Why no S3 measures in outcrop?
  • Lines 241-245, this paragraph should go to the discussion section, not in the results section.
  • Table 2, Fracture set S3: in the intervals in which no discontinuity has been measured, P10 is equal to zero, so to get the average, divide by 6, not by 3 (only the sections in which discontinuities have been found). It would give 0.11 not 0.22.
  • Line 272, the spacing for S1 is the same as the apparent spacing for all families .... this cannot be !!!!!!!!
  • Table 4, it is an unnecessary table as it only repeats the last two columns of Table 2.
  • Lines 288-290, RQD needs to be calculated from the data presented.
  • Line 310, few cm?
  • Lines 312-315, rewrite and explain. It is not understood what the problem is.
  • Line 350, “Except of fracture orientation”, if this is so, then all the previous work is meaningless.

Author Response

Authors answers to the review no. 3:

Dear Reviewer,

Thank You very much for Your effort to revise and improve this manuscript with Your comments and remarks. Here are the answers:

The work presented is a low-cost methodology to measure discontinuities in boreholes. The work may be of interest, but to be published the manuscript must be considerably improved.

First, it is necessary to include in the introduction a more exhaustive bibliographic review of the methodology that is currently used, both commercially and academically, to measure discontinuities within surveys. – noted and improved. This is added: “Recently, digital borehole camera borehole video logging technology (borehole camera, optical and acoustic televiewer) is widely used in exploration technology geological and hydrogeological and nuclear waste research [3–10], Civil Engineering and Geotechnical Engineering [8,9,11–16].”

On the other hand, it is necessary to improve the methodology section. Mainly in the description of the method used. It is not stated where the cameras are located and what method is used to maintain the orientation of the cameras and how the orientation of the discontinuities is determined once the camera is rotated to take measurements. – acknowledged and methodology section is improved. Part added: Borehole camera was transported by terrain vehicle to the drillhole location. Camera was then centered above drillhole by tripod and depth was set to zero m. Camera orientation was checked (on each drillhole) while the camera was still at surface. After all initial checking was set up, the camera was lowered into the drillhole with a winch. Since camera has smaller radius then the drillhole, centering the camera was done with a tripod placed exactly above the drillhole. Camera stillness was partially established by tripod and partially by slow lowering of camera to the drillhole. Camera needs to be still for acquiring precise orientation so sometimes it was necessary to wait for camera to stabilize. Alternatively, it is possible to use stabilizers between camera and drillhole wall but it is dangerous in the open hole but it can be dangerous due to possible collapse of the drillhole wall (in the faulted or karstified intervals).

Subsequently, in the discussion section, what is done is a summary of the results, however, an assessment of the proposed method should be made and the advantages over existing methods should be established. – acknowledged and done.

In this work it is necessary to apply the Terzaghi Weighting (Terzaghi, 1965). Otherwise, the data obtained will have a bias because when orientation measurements are made, a bias is introduced in favor of those features which are perpendicular to the direction of surveying. Without this correction, the data obtained cannot be considered of quality.- Thank You very much for the advice and remark. All data are reinterpreted and Terzaghi bias correction is applied, fractures in each set were revised and the new plots were made in figure 8 as well as statistical parameters in tables.

Finally, there are a series of misprints and errors that need to be corrected:

  • A decimal commas is used throughout the text and must be replaced by a decimal point. - corrected
  • Line 39, Dimension stone cannot be considered as mineral resource. – acknowledged, replaced with natural resource
  • Figure 1, graphic and numerical scale do not match. – acknowledged, numerical scale deleted since scale can be changed during change in text layout in publishing process.
  • Figure 1, In the legend there are no differences in the K1 lithologies – corrected and improved: K2 (1): fossiliferous limestones and brecciated dolomites; K1 (limestones, dolomitic limestones and dolomites).
  • Figure 2A, no graphical scale is placed on a landscape photograph. – graphical scale is lower right corner
  • Lines 137-138, “terrain with no soil or vegetation” is the same as an outcrop. – acknowledged, deleted phrase
  • Figure 6, what are Sd fractures? – Noted that letters are hardly visible on the stereonet. This is improved and new figure is uploaded. It is not Sd fractures it is S0, bedding.
  • Line 209, check english. – corrected and rephased
  • Table 1, there is data in this table that is not understood. For example, why are there only 16 So measures in the survey? Why no S3 measures in outcrop? – Question is understandable! Bedding planes were defined on the drillhole cores based on the angle and morphology of the plane. Since dip angle was around 45 degrees it was hard to say at the time is it the plane actual bedding plane or fracture. For these 16 planes we were certain that they are bedding planes. Why no S3 measurements in the field, we did not find any! Also, S1 and S2 have only few measurements. Surface measurements were good only for bedding planes and that by the sea as in fig 2A. This was the main reason why we came up with this solution of measuring orientation with camera because we would not have practically any measurements. We explain this also in the text.
  • Lines 241-245, this paragraph should go to the discussion section, not in the results section. – noted and moved to Discussion.
  • Table 2, Fracture set S3: in the intervals in which no discontinuity has been measured, P10 is equal to zero, so to get the average, divide by 6, not by 3 (only the sections in which discontinuities have been found). It would give 0.11 not 0.22. – acknowledged and corrected
  • Line 272, the spacing for S1 is the same as the apparent spacing for all families .... this cannot be !!!!!!!! –noted - this is only for drillhole B-3. Depth of the drillhole is 79,4 m and we recorded, 6 fractures of that system in that whole interval. Average apparent spacing for the whole set is 5,67 m. These lines were rephased and only average values were described in text.
  • Table 4, it is an unnecessary table as it only repeats the last two columns of Table 2. Acknowledged. Table is deleted
  • Lines 288-290, RQD needs to be calculated from the data presented. – First parameter that we calculated was RQD. But since RQD is “the percentage of intact core pieces longer than 100 mm in the total” we calculate RQD for all cores between 90-100 % (even in the more fractured parts of deposit). These are logical since we investigate potential dimension stone deposit and if RQD is lower, it would not be the good location for quarry. Based on these conclusions, we moved to P10 and other calculated parameters to define the rock mass quality.
  • Line 310, few cm? – noted and quantified.
  • Lines 312-315, rewrite and explain. It is not understood what the problem is. – we were not able to validate our measurements at this stage of research. We did measure and obtain some preliminary fracture sets but currently no exact way to prove that our measurements were correct in sense of surface cut or underground gallery or similar mining object. This is also logical since our results should give some input to the investors should they proceed with the investment or abandon it. But, by putting our fracture system pattern in the geological structure we can validate our fracture set interpretation, are these sets logical or possible based on the geological structure or not.
  • Line 350, “Except of fracture orientation”, if this is so, then all the previous work is meaningless. – rephased to “In addition to fracture orientation”

Kind regards,

Authors

Round 2

Reviewer 3 Report

  • A decimal commas is used throughout the text and must be replaced by a decimal point. – corrected

They are not all corrected. For example in Fig. 3, in Table 1, in lines 322, 333.

  • Figure 2A, no graphical scale is placed on a landscape photograph. – graphical scale is lower right corner

You cannot put scale, because it is a perspective and the scale changes.

  • On the other hand, it is necessary to improve the methodology section.

It would improve a lot with a photograph of the device with both cameras.

  • Lines 269-273. The data do not match those in Table 1.

  • In the model (Fig. 6, 8) replace anticline axis whit axial plane

  • Lines 415-424 should go to the conclusions section.

Author Response

Dear Reviewer,

Thank You very much for Your effort to revise and improve this manuscript with Your comments and remarks. Here are the answers:

 -  decimal commas is used throughout the text and must be replaced by a decimal point. – corrected They are not all corrected. For example in Fig. 3, in Table 1, in lines 322, 333. – acknowledged and corrected in other places

- Figure 2A, no graphical scale is placed on a landscape photograph. – graphical scale is lower right corner – You cannot put scale, because it is a perspective and the scale changes.- understanded and corrected. Graphical scale removed and in the figure, description is stated that layers are 0,5-1 m thick.

- On the other hand, it is necessary to improve the methodology section. It would improve a lot with a photograph of the device with both cameras. – acknowledged, photograph added to figure 4.

- Lines 269-273. The data do not match those in Table 1. – noticed the error. Fixed.

- In the model (Fig. 6, 8) replace anticline axis whit axial plane - corrected

- Lines 415-424 should go to the conclusions section. - corrected
